# Impact of Coding Educational Programs (CEP) on Digital Media Problematic Use (DMPU) and on Its Relationship with Psychological Dependence and Emotional Dysregulation

**DOI:** 10.3390/ijerph20042983

**Published:** 2023-02-08

**Authors:** Pier Luigi Marconi, Rosamaria Scognamiglio, Elisabetta Marchiori, Daniele Angeloni, Maria Lidia Mascia, Maria Pietronilla Penna

**Affiliations:** 1Artemis Neurosciences, 00184 Rome, Italy; 2Docendum, 00195 Rome, Italy; 3Italian Psychoanalytic Association, 00189 Rome, Italy; 4Department of Pedagogy, Psychology, Philosophy, University of Cagliari, 09123 Cagliari, Italy

**Keywords:** digital education, digital training, Internet use, smartphone use, addiction, emotion regulation, coding educational programs

## Abstract

Alongside the positive effects linked to the introduction of digital technologies into our lives, particular dysfunctional behaviors in the use of digital tools have appeared, through which the expression of conditions such as addiction, difficulties in affective and behavioral self-regulation and mental health problems have been channeled. The present study aims to investigate, in a sample of young students aged m = 12.91 (ds = 0.56) years, whether Coding Educational Programs (CEP), deployed to 44.9% of the sample, is effective in psychological dependence, emotional self-regulation and Digital Media Problematic Use (DMPU), as self-assessed through questionnaires (DERS, DSRS, IAT, MPIQ and MPPUS). CEP had no effect on emotional dysregulation or on DMPU. They were effective in the time management of mobile phone use, with students rescheduling from daytime use on working days to daytime use on the weekend. Moreover, people who attended CEP more frequently used smartphones for orienting themselves and for obtaining information. In conclusion, CEP are effective in achieving a more functional and important use of smartphones and better time management. It is possible that CEP effect on metacognition could reduce DMPU if alternative ways to regulate emotions are available.

## 1. Introduction

The introduction of digital technologies into the modern social, cultural and expressive scenario has supported the emergence of new opportunities for intellectual, personal and creative growth in general [1]. Alongside these positive repercussions, however, particularly problematic behaviors have been detected in the use of digital media, through which conditions of psychological dependence and difficulties in affective and behavioral self-regulation have been expressed [2].

Digital media use can present risks, but, at the same time, they can represent an opportunity for personal, educational and social growth. Digital media have represented a further opportunity to train new generations in the development of their specific cognitive, affective, relational and cultural skills. In their use in primary and secondary schools and in the education context in general, they have contributed to educating students to a constructive and positive use of the new technologies, protecting themselves from risk problems both on a personal and relational level [3]. An example is provided by the use of Coding Educational Programs (CEP) [3].

## 2. Theoretical Framework

The need to question ourselves on digital use and digital education stems mainly from the findings that show how, recently, many young people have developed a problematic use of digital media (DMPU), resulting in the development of actual DMPU-related disorders. This is due to the proliferation of Internet-enabled mobile tools, leading worldwide to serious health and social problems [4,5]. In fact, alongside the positive effects, dysfunctional behaviors have appeared in the use of digital devices, which have also turned into one of the many ways of expressing the psychological and behavioral consequences of young people’s emotional and relational distress [6]. The fact that these tools have now become the main method of communication and contact between young people has made DMPU highly prevalent in new generations and beyond. The impact on the final quality of life is evidenced by a study by Mascia, Agus and Penna [7], where affective dysregulation conditions affect smartphone abuse, which, in turn, impacts everyday functioning.

Numerous studies affirm that DMPU can also lead to addictive behaviors, but this problem becomes much more complex when it is related to the many perspectives potentially present in the use of the smartphone. Undeniably, smartphones can connect to the Internet and also run various types of applications (e.g., games, gambling, social media usage, etc.), the appropriate use of which can increase resources for quality of life; however, the dysfunctional use of them, on the other hand, evidently risks causing a serious psychological and relational deterioration [8,9].

One of the central issues regarding the relationship between dependence (which is a normal psychological attitude in the younger phases of life) and dysfunctional use or abuse of tools or settings is the distinctive definition of addiction or abuse in respect to physiological dependence. John Birtchnell [10], of the University of London, in 1988 attempted to distinguish the concept of physiological dependence, a characteristic of childhood, from addiction that persists into adulthood and is dysfunctional. Therefore, an addicted adult can be seen as “an adult who behaves as if he or she were actually a child”. Addiction is, thus, best understood in terms of the developmental problems from which it derives. These problems support the persistent presence of affective and cognitive incompetencies that lead to a series of incapacities: Successfully separating from the main parental figure and the family as a whole, or from other surrogative relationships;Establishing a secure personal identity by acquiring a general feeling of competence in everyday problems and adequate self-esteem, feeling deserving of adult status like other adults;Possessing adequate self-confidence and adequate trust in the possible solidarity of others.

Consequently, adult dependency is characterized by: The need to be close to others;The inclination to be primarily the recipient in interpersonal transactions;The tendency to relate to others from a position of inferiority and humility.

A dependent person receives from others a borrowed identity, resulting in the emergence of a “false self” [11], a guidance and direction for choices, a compensation for those areas in which they feel incompetent and, above all, acceptance, approval and affirmation of value. These characteristics have been tied by many authors to problematic smartphone use, including addiction. 

The study of the DMPU phenomenon (including abuse and addiction) has its origins in the transfer of the items used in the conceptualization and assessment of substance abuse (such as alcoholism) or compulsive appetitive behavior (such as pathological gaming, compulsive shopping or compulsive eating) in the context of Internet tools.

Thus, past studies have been focused on the similarities of other research on the topic of addiction in other behavioral domains, such as alcoholism.

Using the la Dependence Self-Rating Scale (DSRS) [12], which includes eight of the main components of psychological and emotional dependence (lack of stability, unstructured identity, lack of trust and self-esteem, need for closeness, influenceability, need for protection and care and fear of rejection), Marchiori and colleagues [13] observed that in subjects suffering from alcohol addiction, the level of dependence was only very high when a second diagnosis of personality disorder also emerged; this condition is associated, albeit not significantly compared to the control groups, with a less affectionate and warm parental attitude, and an own attitude of keeping control of what happens in life to oneself. It is also evidenced by conflict between one’s own internal need for external care from and reliance on their family, which tends to the opposite condition, and one’s own cognitive attitude of distrust and hyper-control.

In a 1997 study, Marchiori and colleagues [14] showed that in obese subjects with food appetitive behavior dyscontrol, levels of dependence correlated to the presence of anxious personality traits, but not with dependence on external gratifications, which, on the other hand, tended to correlate to a tendency towards appetitive, impulsive and transgressive behavior.

However, the studies regarding the actual psychological components of DMPU have found quite a more complex framework. DMPU and ICT media addiction, at least, can lead to further harmful behaviors, such as the abuse of the Internet or video games.

Demetrovics and colleagues [15] identified three factors correlated with the level of Internet involvement in one’s life: Obsession, i.e., the level of mental engagement implemented with the Internet;Distraction, i.e., the level of distraction from daily activities, relationships and essential needs;Dyscontrol, referring to difficulties in controlling Internet use [15].

Billieux [16] summarized the current open lines of enquiry, indicating four groups in the research on problematic mobile phone use (today, we talk about smartphones): Impulsivity; limited capacity for self-control and emotional regulation;Relationship maintenance, which portrays cell phone abuse as a means of obtaining security in emotional relationships and is characterized by low self-esteem and high levels of neuroticism;Extroversion, which associates excessive use with sociality and an intense desire to maintain relationships;Cyber addiction in consonance with mobile phone technology, which allows access to various online utilities and applications and sustains misuse as a result of the attraction of this technological environment.

The evidence of emotional and behavioral dyscontrol has highlighted the importance of emotional dysregulation in the development of addictions and of parental functioning during childhood in the development of an adequate emotional regulation in adults. The appropriate relationship of emotional dyscontrol with pathological dependence or abuse and addiction has become the central issue of the most recent studies.

Gioia and colleagues [17] published a review about DMPU and emotional dysregulation. They reviewed 23 studies from 2010 to 2020. The considered studies supported the role of emotional regulation deficits in the development of problematic Internet use, which can be considered a compensatory coping strategy for managing emotional overload [17]. 

Moreover, they found data supporting the role of metacognition in the relationship between emotional dysregulation and DMPU. In fact, this relationship was found to be mediated by positive metacognitions (such as escapism and controllability) [18,19] and cognitive factors (such as experiential avoidance and desire thinking) [20]. However, having good metacognitive skills does not protect at all from problematic ICT use. In fact, as 2016 Casale [19] suggested and Gioia and colleagues affirm from the evidence of the considered studies, “when alternative and more adaptive emotional regulatory strategies are not available, individuals might turn to the Internet in order to escape from reality and get greater control over negative thoughts and feelings compared to real life situations”. 

### 2.1. Risk Factors

The interpretative model adopted in studies about DMPU (and smartphone abuse in particular) makes reference to the framework of psychological dependence, even if, as mentioned above, in recent years, the role of emotional dysregulation and metacognition becomes more evident.

#### 2.1.1. Demographics

Many studies were conducted only describing the simple association between demographic data and problematic ICT use.

The most widely documented risk factor is being male. The prevalence among boys and young men has been found to be 2 [21], 3 [22], or even 5 [23] times higher than among girls and young women. During early adolescence, problematic use increases with age, but peaks around 15–16 years of age [24].

#### 2.1.2. Social Context and Parental Functioning

The importance of the social context and the quality of parental functioning in relation to the development of problems in smartphone use has also been highlighted.

Indicators of lower socio-economic status, including lower maternal education and a single parent family, have been shown to increase the risk of the emergence of such behavior [25]. Several studies evidenced the relationship between Internet abuse and relationship problems in the family. Risk factors appear to include lower levels of family cohesion, family conflict and poorer family relationships [26]. Sanchez Martinez and Otero [27] confirmed a relationship between students and problematic mobile phone use, negative family relationships and highly educated parents without financial difficulties. They explain that this relationship is due to the need to maintain compensatory social relationships. In terms of family upbringing, Zhou and colleagues [28] also observed a significant relationship between parental abuse and dependence on mobile phones and children’s dependence on the Internet and other technologies, which they interpreted as the result of emotional neglect.

In a 2014 study [29], it was observed how the presence of anxiety depressive disorders, but also of a dysfunctional family and friendship social network, can condition mobile phone overuse. 

#### 2.1.3. Psychological Distress and Dysfunctions in Emotion and Behavioral Control

The reported risk factors described above are all coherent with the found relationship between emotional dysregulation and problematic ICT use.

The presence of psychological distress was also observed in many studies on the relationship between mobile phone (smartphone) abuse and personality traits.

Dysregulation and appetitive novelty seeking are psychopathological dimensions confirmed as risk factors in compulsive behaviors such as gaming. 

Billieux and colleagues [30] found that urgency, lack of forethought and lack of persistence are inversely correlated with self-control. 

However, urgency, defined as the tendency to experience strong urges that cannot be postponed due to negative affective states, is the component that best predicts DMPU. Thus, a high urgency score relates to a higher number of calls and duration and number of text messages sent. Urgency is similarly related to inadequate strategies for self-regulating negative emotional states. Billieux [16] proposes that urgency can derive from two factors: Lack of perseverance, which may be reflected in the number and duration of phone calls, as well as in associated financial problems;Lack of forethought, which may lead to their use in dangerous or forbidden situations, which are linked to sensation seeking.

Sensation seeking is a personality trait involving the dimensions of thrill and adventure seeking, lack of inhibition, experience seeking and sensitivity to boredom [31,32]. It is characterized by the need for new experiences that are unusual, varied and intense, with accompanying physical, social, legal and/or financial risks, often coexisting with impulsivity in addictive behavior.

This trait can be related also to the “desire thinking” highlighted by Gioia [17] as cognitive mediators between emotional dysregulation and problematic Internet use.

Previous studies have found a relationship between leisure boredom and self-esteem; Leung [33] confirmed that boredom as measured by the Leisure Boredom Scale [33], sensation seeking using the Adventure subscale [32] and self-esteem via the Rosenberg Self-esteem Scale [34] are significant predictors of DMPU. 

Many papers have been published on the many of the components of affective dysregulation, such as impulsivity. High levels of impulsivity, as well as a limitation of self-control, are found in users who abuse Internet gaming. In two studies examining problematic smartphone use, one identified dysfunctional impulsivity and low self-control as two key risk factors [35] and the other found impulsivity to predict this behavior in their female participants [36]. Patients diagnosed with a gaming disorder also demonstrated higher levels of impulsivity than healthy controls [37]. A systematic review of research examining personality traits predictive of DMPU concluded that impulsivity plays a role in abuse of Internet gaming and that certain aspects of this trait, such as high levels of urgency, are particularly powerful risk factors [38]. 

In addition to impulsivity, behavioral traits related to aggression and hostility are common among adolescents with media use disorders. Aggressive tendencies were identified as a predictor of impulsive and additive gaming by multiple studies in a recent research review [38]. Aggressive tendencies were identified as a predictor of gaming dependence by multiple studies in a recent research review [38]. In a large European survey study, adolescents who reported addictive gaming on the Internet had higher scores on the rule-breaking and aggressive behavior scales [39]. Although it may appear that aggression scores are simply indicative of observed gender differences, models including gender and other traits predicting DMPU found that hostility was independently associated with smartphone abuse [40], and conduct problems were predictive of problematic Internet use [41]. Walsh and colleagues [42], by means of the Mobile Phone Involvement Questionnaire (MPIQ), differentiated the frequency of mobile phone use for personal application from its use due to addiction. From observed data, authors affirmed that a difference must be made between “involvement” and “frequency” in the use of smartphones: Self-identity, to which belongs the perceived value of the mobile phone for self-concept and to relate with others, is a predictor of frequency of use;Self-identity and the approval of others determines instead dependence or high involvement with smartphones.

That is, they considered dependence on mobile phones to be correlated with dependence on social environment, as well.

### 2.2. Protective Factors

Medical and educational institutions, governments and other groups have sought to take preventive action or to treat disorders related to DMPU. 

In many cases, the preferred treatment for these disorders is to establish recommendations for appropriate use. 

Psychosocial therapies (including cognitive behavioral therapy, family therapy and compounded therapy) for these DMPU-related disorders and pharmacotherapies (including antidepressants and psychostimulants) for comorbid psychiatric or developmental disorders have reportedly been effective in reducing the degree and symptoms of DMPU. 

Positive parent–child relationships may be protective against the development of problematic gaming [43]. Furthermore, the parental monitoring of adolescents’ Internet use may also reduce the risk of DMPU, which, in turn, improves parent–child relationships [44]. Parents, it seems, have some prevention tools at their disposal that could improve their overall family functioning. Fathers seem to have a particularly influential role, as their relationships with adolescents have proven to be particularly protective [43,44]. Some risk factors for DMPU (e.g., being male, suffering from attention deficit hyperactivity disorder and showing worsening psychiatric symptoms) have started to be identified. 

However, clinical studies, treatment and preventive actions are insufficient for the treatment of these problems, and standard treatments and preventive systems have yet to be established. 

Educational and medical institutions, government, families and others need to take more action and cooperate more effectively to treat or to prevent DMPU-related disorders.

In some countries, therapeutic camps have been developed for adolescents with DMPU-related disorders, and preventive education (including lectures and group discussions) has been provided for general adolescents. Such efforts have been effective in reducing the average degree of the severity of such disorders. 

Their use in primary and secondary schools has represented a further opportunity to educate the new generations, both in the development of their specific cognitive, affective, relational and general cultural skills and in educating them in the constructive and positive use of new technologies, protecting them from risk issues both on a personal and relational level. 

An educational activity that has become increasingly widespread in Italy in recent years is training in digital citizenship, and courses supporting the development of cognitive, relational and affective skills, with education in the logical programming of small games up to the programming of real robots (CEP).

### 2.3. Aim and Hypotheses

On the basis of the literature taken into account, the following hypotheses have guided our study. This work aims to reach three goals:To investigate the relationship between the experiences of affective dependency (both in general and in its sub-components), affective dysregulation and DMPU;To study how participation in CEP can influence these relationships;To study the behavioral effect of CEP on the proper use of digital media and Internet tools.

## 3. Materials and Methods

### 3.1. Participants

Participants were all students from six different classrooms in two middle schools in Rome. They were recruited as volunteers, after a proposal to their teachers and parents. One school was private and the other public. The economic and cultural backgrounds of the families were similar, but the urbanistic area of the city was different, even if many of the available facilities were similar.

### 3.2. Procedures

#### 3.2.1. Coding Education Procedure

Coding was introduced in school teaching to implement the development of computational thinking through the use of digital tools and computer languages. It stimulates, by means of gaming contexts for younger students, those mental processes that use logic for algorithms for the purpose of solving a problem by breaking it down into sequential steps [45]. This process facilitates and makes the achievement of the objective more effective. In this learning environment, logical processes using sequences, repeating loops, choices “if/then”, abstraction, debugging and generalization are used [46]. Coding also includes the exercise of social competences focused on the value of collaboration. It also involves the creative approach of thinking to find new solutions starting from a problem [46]. Teaching students the formative field of coding has a fundamental objective; not only through the acquisition of procedures but also through the use of models, it plays a role in orienting students towards the outside world with a greater programmatic awareness of organization in daily life.

#### 3.2.2. Questionnaires Administration

The administration took place through the distribution of the questionnaires in the form of paper, filled in by the participants inside the school classrooms, between 9 and 13 h on an ordinary morning at school. 

The average compilation time for each class was about 25′. 

The participants compiled in silence, in the presence of the curriculum teacher.

### 3.3. Measures

The sample of children participating in this study who reported having attended coding courses followed a structured path in extracurricular hours since the early years of primary school. There was one hour a week of training for a total of about 20 h a year. Later in middle school, many of the participants attended digital citizenship courses in parallel with coding. 

The learning of coding has followed the international lines of teaching computational thinking through open-source platforms specifically designed for schools in a gaming approach, in use in Europe since 2014, according to the method “Blockly” (see scratch.mit.edu and code.org, Minecraft Education Edition). They also use methodologies of tinkering, orientation in physical and virtual space, storytelling and coding unplugged.

With regard to digital citizenship, participants addressed the concepts of risks, rules and opportunities of the web in a tech lab environment.

#### 3.3.1. Dependence Self-Rating Scale (DSRS)

The Dependence Self-Rating Scale (DSRS) [47] is a self-rating scale of psychological and emotional dependence consisting of 48 items selected from several already validated measurement scales. The DSRS assesses eight of the main components of psychological and emotional dependence as described in the literature [10]: lack of stability, unstructured identity, lack of trust and self-esteem, need for closeness, influenceability, need for protection and care and fear of rejection. Subjects are asked to answer direct questions concerning thoughts, feelings and behavior related to these components. The items are evaluated by a Likert scale (from 1 to 4): never, sometimes, often, always. Validation studies [13,47] indicate that it has good internal consistency and has been used in subsequent studies for the assessment of psychological and emotional dependence related to personality traits and interpersonal relationship patterns [13]. 

#### 3.3.2. Difficulties in Emotion Regulation Scale (DERS)

The Difficulties in Emotion Regulation Scale (DERS) [48] is a self-report scale created to measure emotion regulation problems. This tool consists of 36 items that produce scores on the following six subscales: nonacceptance of emotional responses; difficulty engaging in goal-directed behavior; impulse control difficulties; lack of emotional awareness; limited access to emotion regulation strategies and lack of emotional clarity. The Difficulties in Emotion Regulation Scale (DERS) is one of the tests used most frequently to assess the difficulties in emotion regulation among a population; the DERS can be considered a useful tool to measure emotion regulation strategies in the Italian context as well [49].

#### 3.3.3. Internet Addiction Test (IAT)

The Internet Addiction Test (IAT) [50] is a scale that consists of 20 items created to assess the relation between Internet use and the respondent’s productivity at work, school or home (3 items), social behaviors (3 items), emotional connection to and response from using the Internet (7 items) and general patterns of Internet use (7 items). The items are evaluated by a Likert scale from 1 to 5 (from “does not apply” to “always”). This scale has been validated in many countries around the world.

#### 3.3.4. Mobile Phone Involvement Questionnaire (MPIQ)

The Mobile Phone Involvement Questionnaire (MPIQ) [42] is a self-report scale to measure the cognitive and behavioral association with mobile phone use. The scale is made up of eight items created in order to assess withdrawal, cognitive and behavioral salience, euphoria, loss of control, relapse and reinstatement, conflict with other activities and interpersonal conflict. The items are evaluated on a Likert scale from 1 (strongly disagree) to 7 (strongly agree). Based on the criterion of a reliability coefficient of 0.70 or higher being considered acceptable, the scale was reliable (α = 0.78).

#### 3.3.5. MPPUS

The Mobile Phone Problematic Use Scale (MPPUS) [51] is a self-report scale developed to measure the problematic use of mobile phones (PSU). In the literature, we can find many validations of this scale around the world. The Italian version of the scale [52] is composed of 24 items divided in two subscales covering the issues of tolerance, escape from other problems and craving and withdrawal and negative life consequence in the areas of social, familial, work and financial difficulties. All items are measured on a scale ranging from “not true at all” (1) to “extremely true” (5). The higher the score, the more problematic the use of the smartphone. The questionnaire has reported a Cronbach’s alpha of 0.93, demonstrating a high level of internal consistency. 

### 3.4. Data Analyses

The statistical analyses were carried out by the application IBM SPSS. The evaluation of the level of statistical significance of the association between nominal variables (sex, school, attendance of the Coding Educational Program and finalities for which smartphone is used) was carried out by the Chi squared test. The statistical significance of differences in values of parametric variables (time spent with the smartphone, age, psychological dimensions, mobile phone involvement and problematic use) was performed with Anova1Way. The discriminating variable was attendance of the Coding Education Program. 

To reduce original parameters to root factors, which can also be interpreted as background dimensions assessed by all the instruments, a principal component factorial analysis was performed. The number of factors was determined by scree plot as visual criteria and eigenvalue higher than 1 as computational criteria. The final solution was found after varimax rotation.

The association between parametric variables was computed by bivariate Pearson r. Bivariate correlation tables and Pearson r computation was conducted to better interpret the meaning of the factors and the dimensional composition of the original variables.

Correlations between extracted factors were computed to study the linkage between psychological variables (assessing emotional dependence and dysregulation) and the variable assessing behavior and mental attitude to mobile phones. The two sets of extracted factors were intercorrelated. This study of association was performed three times: on the whole group of young people, among the data of those who attended the Coding Educational Program and among the data of young people who did not attend.

A discriminant analysis was performed to find the best function discriminating between people attending Coding Educational Programs and those not. The forward method was adopted, with *p* < 0.05 to include and *p* > 0.10 to exclude from the function.

## 4. Results

### 4.1. Characteristics of the Observed Population

98 students (56 male and 42 female) participated in the study. They were on average m = 12.9 (sd = 0.57) yrs old. The participants were attending the final academic year just before High School. 56 of them (57%) had already attended an educational program on digital culture at m = 9.4 (sd = 2.5) yrs old, and 44 (44.9%) declared to have been participating in CEP (see Table 1). 

The prevalent use of smartphones among participants (see Table 2) was for chatting and accessing social networks (91.5%). Only 60.6% used smartphones for phoning. Only about one half used a smartphone for obtaining information (46.8%) or gaming (51.1%). The last use is the only one in which sex differences were detected (M = 58.5% and F = 34.1%; *p* < 0.05). Only 12.8% used a smartphone to help with orientation.

The participants also declared the time they spent on mobile phones (Table 3): m = 8.0 (sd = 7.3) hours per week during the day and m = 1.8 (sd = 3.1) hours per week at night during working periods, while during weekends they spent m = 5.9 (sd = 3.8) during the day and m = 2.4 (sd = 4.3) at night.

Among the whole group, 44 students (44.9%) attended CEP. The difference between the two schools in number of students who have already attended CEP was significantly different (X^2^ = 16.498; *p* < 0.001) (Table 1).

### 4.2. Effect of Coding Educational Program on Mobile Phone Problematic Use

No evidence of a CEP effect was detected on any of the instruments used to assess DMPU (see Table 4).

### 4.3. Effect of Coding Educational Program on Self-Regulation and Affective Dependence

Rating scales assessing emotional difficulties either in emotional self-regulation (DERS—Table 5) or in affective dependence (DSRS—Table 6) evidenced no differences due to CEP. One subscale of DERS (Attention Impairment) showed a significant sex effect (M = 13.0 + −3.8; F = 15.6 + −4.8; *p* < 0.01), as did a subscale of DSRS (Being Cared Self Identity: M = 5.6 + −0.9; F = 6.2 + −1.0; *p* < 0.01). Non-differences were detected among students of the two different schools, but one DSRS subscale (Self-Sufficiency) showed differences (Private = 3.8 + −0.8; Public = 3.4 + −0.8) that resulted statistically significant (*p* < 0.05).

### 4.4. Psychological Dimensions of the Affective Dysregulation and Affective Dependence

The factor analysis performed on the original 10 variables used to assess the personal style of emotion management resulted in the extraction of the independent factors (Table 7).

#### 4.4.1. First Factor of Affective Dependence and Dysregulation: Emotional Dysregulation

The first factor (Emotional Dysregulation), explaining 36.7% of the observed variance, is focused mainly on the personal difficulties in emotion management. Five of the six subscores of DERS depend on it. Only the Reduced Self-Awareness subscore of DERS, focused on the self-perception of inner feeling, is excluded from this factor. At the same time, the DSRS subscale “Being Cared Self Identity” relies on this main factor. This factor seems to assess the core component of DERS.

#### 4.4.2. Second Factor of Affective Dependence and Dysregulation: Reduced Self-Confidence

The second factor (Reduced Self-Confidence), explaining 17.4% of the observed variance, includes two of four DSRS subscores (Lack of Self-Confidence and Being Cared Self Identity) and one DERS subscale (Reduced Self-Awareness). This factor seems to assess the lack of trust in both others’ and one’s own capabilities to resolve emotional overload, and at the same time is increased by a missing awareness of inner feeling.

#### 4.4.3. Third Factor of Affective Dependence and Dysregulation: Self-Sufficiency

The third factor (Self-Sufficiency) includes the last two subscores of DSRS (Complaisance and Self-Sufficiency), which express the dimension of the dependent attitude assessed by DSRS not also intercepted by the DERS.

In Table 8, the correlations between the three factors and the Total Score of the two questionnaires, DERS and DSRS, are shown. It is evident that the first factor extracted expresses the assessment of the DERS, while the DSRS is split in two components. The factor “Reduced Self-Competence” seems to express features not identified by both DSRS and DERS, even if it is computed by three of their subscores.

### 4.5. The Dimensions of Mobile Phone Problematic Use

A factorial analysis was performed on the parameters assessing the behavior linked to mobile phone use. Even in this case, three factors were extracted (Table 9).

#### 4.5.1. First Factor of Problematic ICT Use: Problematic Use

All the questionnaires assessing problematic attitudes toward mobile phone use rely on the first factor (Problematic Use), explaining 39.5% of the variance. This means that all of them assess the same phenomena (the problematic use of ICT in general and mobile phones in particular), and they could provide an actual assessment of the DMPU dimension.

#### 4.5.2. Second Factor of problematic ICT Use: Night Use

The second factor (Night Use), statistically independent from the first one, refers to the time spent using the smartphone at night, either during working days or on the weekend. It explains 20.2% of the observed variance.

#### 4.5.3. Third Factor of problematic ICT Use: Day Use

The day use of smartphones is loaded in the third and last factor (Day Use, explaining 20.1% of observed variance).

These three factors split in three different criteria the describe of the behavior associated with a problematic use of new media devices. The results highlight the differences between the presence of problematic attitudes toward the use of these ICT media and the time schedule adopted for accessing these instruments.

In Table 10, the correlations of the original parameters with the three extracted factors are shown. The data confirm the interpretation of the three factors, also highlighting the inclusion of time spent with the smartphone assessed by the MPIQ scale.

### 4.6. The Relationships between the Dimensions of Psychological Distress and the Dimensions of Problematic Use Behavior

In Table 11, the correlation between the two sets of factors (Psychological Affective Factors and Behavioral Factors) are shown.

It is evident that DMPU (Problematic Use of smartphones) is strongly associated (r = 0.574; *p* < 0.001) with the observed degree of Emotional Dysregulation. There is no evidence about the influence of Reduced Self-Confidence on DMPU-related observed behaviors. A weak link (r = 0.320; *p* < 0.05) is instead shown between the level of Self-Sufficiency and the Night Use of smartphones. Having attended coding educational programs does not seem to significantly modify such relationships, but has a possible weak effect on the relationship between Night Use and Self-Sufficiency.

The regressive function to predict the level of DMPU (factor FB1—DMPU—Problematic Use) on the basis of the other extracted factors, computed including one variable per time, yielded significant results only including the factor FA1—Emotional Dysregulation (see Table 12). This result confirms the fundamental correlation between DMPU and the presence of emotional dysregulation.

The same procedure repeated including only participants who attended CEP or not yielded similar results, without any apparent effect due to having attended CEP.

### 4.7. The Difference Linked to Having Attended a Coding Educational Program (CEP)

The discriminant analysis, performed on all the sets of data (affective parameters, behavioral parameters and extracted factors) on the basis of having attended a CEP, yielded significant results only regarding the exchange of the timetable of smartphone usage, with a decrease in time spent during the day in working days and a shift toward the use in the weekend (Table 13).

On the other side, a significant linkage was found between attendance to CEP and the use of smartphones to obtain information or to help with orientation.

## 5. Discussion

### 5.1. Problematic ICT Device Use: A Unique Dimension?

The first evidence we obtained from this study is about the root dimension in common with all the different behavioral correlations of DMPU. In fact, ICT media were developed to increase tools and resources for managing everyday activities. The dysfunctionality rises when such a goal is not reached and the ICT resources become a cause of dysfunction in everyday life: being late in reaching goals, focusing attention and energies in something else that has no impact on everyday life and founding interpersonal relationships on something which is highly volatile and at risk of not being true. The behaviors that we detected in people which express such dysfunctionalities seem to be correlated with each other, as they were expressions of the same psychological and psychopathological phenomena. 

The many assessment instruments, proposed throughout time by different authors, seem to be differentiated by the specific ICT tools focused on at that time, as well as the ICT tools more accessed in that time by people. However, they seem not to be expressions of actual different psychological dynamics. In the 1990s, at the end of the last century, the Internet was the new available resource, accessed with a personal computer. The next decade increased the functionalities of mobile phones and their use to access the many Internet resources. In changing and increasing their functions in the following years, their names were also changed from mobile phones to smartphones. The programs accessed were, at the beginning, more text-based than multimedia contents. Now, the multimedia contents have become the most used functions, especially by young people. The different assessment instruments we used in this study refer to Internet use, mobile phone use, and ICT resource use in general. The evidence we obtained here is that, as the different instruments rely on the same main factor, the different behaviors to which they make reference are all expressions of the same psychopathological dimension: an inappropriate use of ICT media beyond its original purpose. With this concept, we adopted in this paper the general term DMPU—Digital Media Problematic Use—as the common dysfunctional dimension.

### 5.2. Time Scheduling Has Something Different from the Dysfunctional ICT Use?

The second important result evidenced by this study is about the functional role of time management regarding the use of ICT devices. Many assessing tools include time scheduling as a construct of DMPU. The MPIQ, for example, includes time management in the total score, which is also confirmed by correlations shown in Table 11. However, time spent at night and time spent during the day affect two different factors. This can be interpreted as time scheduling being independent from the level of problematic use and that it could be a difference in factors leading to night use from factors leading to the use of smartphones during the day. Already Walsh and colleagues [42] have proposed that “frequency of use” and “involvement” in DMPU could be linked differently to two factors (personal identity and social approval). In our result, having attended CEP in the past leads students to express a different daytime planning of their activities, at least in the use of smartphones. Actually, the intervention of digital culture, in different ways according to the type of intervention performed, also operates by increasing the skill of active time management. Teaching students to program the sequence of actions, CEP also teaches managing time in everyday life, or, at least, the reported times spent with the tool. Therefore, the relative ratio is changed between time spent with smartphones during working days and time spent on the weekend. On the other hand, time spent at night seems to be related (Table 12) to the feeling of higher self-sufficiency, that is, a further concept of independence. It means that nighttime activities are planned in relation to a feeling of not needing the protection of others, and, consequently, being compliant with them. This hypothesis can also drive the consequent further hypothesis that external ruling at night is not working in young people who feel able to face non-protection by parents themselves and, thus, to decide by themselves about their own life. Therefore, this kind of feeling could drive young people to find ways to act outside parental control, such as at night. However, even with this aspect, Table 12 suggests the possibility that having attended CEP could drive a change in the relationship between self-sufficiency and night use (people who attended CEP show no correlation between the two factors). This could be due to an increase in awareness in CEP students, together with their increase in capability for time management. A higher competence in coding can induce a higher competence in orientation, planning and responsible decision making. Casale [19] and then Gioia [17] have affirmed the importance of metacognition in the modulation of the relationship between emotional dysregulation and problematic ICT device use. The effect on timing could be an expression of an increased skill, through coding education, in obtaining the appropriate meta orienting needed to increase awareness about everyday life management, even if the correlation between emotional dysregulation and DMPU is not changed. As Gioia [17] affirmed in their review paper, to find an impact on this relationship, however, actual effective instruments to regulate emotions must be available.

### 5.3. Which Is the Risk: Psychological Dependence or Emotional Dysregulation or Both?

The third evidence is about the psychological dimension linked to DMPU. Most of the present literature frequently refers to emotional risk factors as psychological discomfort and the presence of affective dependence. Even if the behavioral and personality traits, detected as prevalent in people with DMPU, suggest the importance of emotional dysregulation, only more recent studies are highlighting the importance of this dimension. 

In this study, we have matched two different scales to assess the emotional problems that could be linked to DMPU, one focused on psychological dependence (DSRS) and the other on emotional dysregulation (DERS). 

The different psychological constructs between these two scales are confirmed by the factor analysis, which extracted three different factors. Two of these factors appear to be formed differently by DERS and DSRS subscores. The first one (Emotional Dysregulation) is more likely to be an expression of Emotional Dysregulation, while the other (Self-Sufficiency) refers to psychological dependence, even if the psychological construct of DSRS seems to include different dimensions affecting all the three factors extracted. Only one of the three factors (the Emotional Dysregulation factor) is related to DMPU. This observation sustains a change in the original model of dependance, splitting physiological and pathological dependence. The addiction was correlated with the latter. Recent literature, however, and the present study, confirm dysregulation is the main correlation of addiction, abuse and dysfunctional use in general. As such, this means that these dysfunctional behaviors are spontaneous attempts to manage emotional overload with external actions, when one finds oneself incompetent in adopting inner actions, which are needed to self-regulate. These data are coherent with the more recent literature, in which the emotional and behavioral components of emotional dysregulation were found to be a risk factor for DMPU. 

As further evidence, moreover, a third dimension was found (Self-Confidence), related to the lack of trust either in themselves or in others, together with a lack of clear awareness of one’s own feelings. 

In other words, our data support a three-component model of dependence: Lack/presence of competence in self-regulation of emotions (Emotional Self-Regulation);The persistence/release of a strong affective attachment and compliant attitude towards caregivers (Self-Sufficiency);The presence/absence of a trusting attitude toward self and others associated with a sufficient level of consciousness of inner feelings (Self-Confidence).

The third component (the second factor extracted in this study) as an independent dimension form the other two must be further studied, especially its meaning and the reason why it is separated from the other two. The Emotional Dysregulation factor seems to be the only one linked to the problematic use of new ICT resources, while Self-Sufficiency seems to be linked to the use of ICT devices outside the ruling control of adults. Instead, the Self-Confidence factor remains uncorrelated with anything. 

It could be linked to the concept of hopelessness and helplessness, a psychopathological component which makes this factor as a measure of the risk of young people to be on the edge of despair. In turn, such a feeling is an important risk factor to dramatic behavior when the social net could be suddenly broken by an unexpected event with high negative valence. 

### 5.4. What Is the Actual Impact of Coding Educational Programs on DMPU?

The recent literature suggests that the role of families in sustaining a better self-regulation and ruling is adopting more functional planning in everyday activities. The educational programs delivered in school could be useful in sustaining a more functional use of ICT devices and increasing the awareness of young people about them, located in the tools/time/space/goals integration as it is done in planning and programming. CESP has all these prerequisites to be an effective educational program for this purpose.

#### 5.4.1. Efficacy on More Functional ICT Device Use by Teaching Students More Useful Applications and Tools Accessible through Smartphones

Data shown in Table 14 highlight an extended number of Internet resources used, available through smartphones, detected in young people who attended CEP. Young people taught and trained on coding affirmed more frequently than those who do not that they used digital devices as instruments to obtain more information and orientation, extending all their potentiality as an information device. Thus, these young people are helped by increasing their competence and knowledge about ICT resources and their strategic use in everyday life and in obtaining more information by digital media; this wider use of digital devices in these young people balance their use better, rather than treating them only for social media (this use, of course, is continued) and/or an emotion manager tool.

#### 5.4.2. Efficacy on More Functional ICT Device Use by Teaching Students More Functional and Aware Time Management of Their Activities

Moreover, educational programs make young people more aware of the time management of their activity, even in smartphone usage. In any case, the presence of personalities with higher feelings of self-sufficiency makes them free to manage as they like the “night” time, where no duty is present (for example, getting in touch with friends and gamers by smartphone). However, even in such self-sufficient people, a CEP effect is detected. This “time planning” effect could be related to the role taken by CEP in improving the orientation of young people. This increased skill could also have an effect in their competence to put themselves in a “third position” looking at their own life among others and the personal aims in the life. This means they increase their capability to see the present time scheduling and personal relationship scenario from a position from where they see themselves and other from an upper view. Coding can support such a skill, as all the procedures and different choices must be overviewed and programmed in a sequential algorithm in time and space domains. From the capability of seeing an overview from a “third position”, metacognition about oneself and others can be sustained. The role of metacognition was already proposed as a modulator of the relationship between emotional dysregulation and problematic ICT device use.

#### 5.4.3. Lack of Efficacy on More Functional ICT Device Use by Increasing Emotional Self-Regulation

It is evident that educational programs do not affect emotional dysregulation, and in this way, they do not affect the DMPU attitude of young people. In spite, the fact is that CEP increases the skills and competence of young people in using ICT resources more efficiently and in managing their everyday life activities This is not enough to resolve the effect of emotional dysregulation on DMPU. The facilitation of changing such relationships can be done by better competence on functional use and better daytime planning with an increased metacognition and awareness, if the young people have different and more functional tools and ways to manage emotions.

#### 5.4.4. The Educational Effect of Coding

CEP are effective in increasing the competence of participants in the direction of the educational objectives expected: acquisition of a way of thinking characterized by greater logical abilities and procedural skills, in parallel with a greater capability of the subjects to orient themselves in the external world—more than in the internal—to support the improvement in “problem analysis and organization towards solution” in everyday areas. However, these are not Educational Programs on internal emotion management and regulation. The availability of adequate resources for young people to reach better emotional management makes students more probable to ameliorate the dysfunctional use of digital resources, while the lack of such resources and correlated skills makes the addictive behavior more compulsory. Table 12 is coherent with this interpretative hypothesis.

## 6. Conclusions

Our results confirm the need of implementing educational actions based on programs promoting digital awareness leading to a more functional and regulated use of digital tools. In the present study, the importance and effectiveness of CEP is observed mainly on the behavioral level. An impact of training on emotional self-regulation is not observed. Could be this a question of time? Further studies with higher number of participants are needed. With regard to psychological dependence, since we are dealing with developmental age subjects, it is not possible to assess in terms of stable personality traits, but Table 8 suggest the correlation between emotional dysregulation and a component of dependence (a self-image to be protected). This may suggest that the presence of a lack of competence in the self-regulation of emotions could lead to a persistence of referring to others and to the external world (with higher risk in developing addiction) as a way of regulating emotions. However, the evidence of the modulating power of metacognition on additive behavior in the presence of available alternative resources for self-regulation leads us to highlight the importance of early CEP taught constantly in a longitudinal way. In this perspective, the metacognition programs included in CEP have a strategic importance, being designed with the goal of stimulating orientation, especially the visual–spatial part, which is helpful in everyday thinking and reasoning by objectives and cycles. It would also be interesting to know the effect of Educational Programs on Digital Citizenship (DCEP) on emotional regulation and on the quality of social relationships. Further studies are needed with more appropriate samples of students. The DCEP represent an opportunity to strengthening the relationship with students, consolidating a sense of trust in sharing personal experiences as well. Dealing with expert trainers, external to school faculty, considered by student competence on programming strategic behavior, can help them adopt more responsible behavior. This aspect leads us to reflect on how the concept of norm has been changed over time., In the past, the norm had a transcendent and compulsory value. Today, the norm has no force in being implemented because it is not proposed and certified by a trusted authority. In this regard, programs such as CEP (or DCEP) take on a normative value, by an operational and pragmatic effectiveness. For younger students, the game context, proposed by current CEP online platforms, significantly helps students get involved in this normative perspective. The limitations of the research include the small sample size and the use of self-report instruments.

## Figures and Tables

**Table 1 ijerph-20-02983-t001:** Description of school, sex and age of participants.

School	Participant	Age (yrs) m (ds)	M	F	CEP
Public School	49	12.81 (0.39)	29 (59%) ^1^	20 (41%) ^1^	12 (24.5%) ^1^
Private School	49	13.02 (0.69)	27 (55%) ^1^	22 (45%) ^1^	32 (65.3%) ^1^
Total	98	12.91 (0.56)	56 (57.1%) ^2^	42 (42.9%) ^2^	44 (44.9%) ^2^

^1^ Percent of the total number of that school participants. ^2^ Percent of all 98 participants. CEP: Coding Educational Program.

**Table 2 ijerph-20-02983-t002:** Prevalent use of mobile phones among participants.

Sex (N)	Phoning	Chatting and Socials	Gaming	Obtaining Information	Getting Oriented
M (56)	28 (52.8%) ^1^	48 (90.6%) ^1^	31 (58.5%) ^1^ *	26 (49.1%)	6 (11.3%)
F (42)	29 (70.7%) ^1^	38 (92.7%) ^1^	14 (34.1%) ^1^ *	18 (43.9%)	6 (14.6%)
Total (98)	57 (60.6%) ^2^	86 (91.5%) ^2^	56 (57.1%) ^2^	44 (46.8%) ^2^	12 (12.8%) ^2^

^1^ Percent of subjects among the same sex group. ^2^ Percent of subjects among the whole group. * Significant sex effect (*p* < 0.05).

**Table 3 ijerph-20-02983-t003:** Time spent using mobile phones during the day or night, in working days or weekends.

Sex	N	Day Use Working Days m (ds) ^1^	Day Use Weekend m (ds) ^1^	Night Use Working Days m (ds) ^1^	Night Use Weekend m (ds) ^1^
M	56	7.7 (7.0)	5.2 (3.5) *	1.9 (3.3)	2.2 (4.2)
F	41	8.4 (7.6)	6.8 (4.1) *	1.7 (2.7)	2.7 (4.5)
Total	97	8.0 (7.3)	5.9 (3.8)	1.8 (3.1)	2.4 (4.3)

^1^ Total hours in the whole period. * Significant sex effect (*p* < 0.05).

**Table 4 ijerph-20-02983-t004:** Mean and standard deviation of the variables used to assess DMPU and their differences by sex, school and participation in CEP.

Problematic ICT Use Assessment	Sex		School		CEP	
IAT—Total Score ^§^	M = 56	47.0 (9.6)	Private = 49	44.8 (11.6)	Yes = 44	45.9 (11.4)
	F = 42	44.3 (12.0)	Public = 49	46.9 (9.7)	No = 54	45.8 (10.2)
	Total = 98	45.8 (10.7)	Total = 98	45.8 (10.7)	Total = 98	45.8 (10.7)
MPIQ—Total Score ^§^	M = 56	26.3 (8.0)	Private = 49	26.1 (9.7)	Yes = 44	26.6 (10.4)
	F = 42	28.4 (10.0)	Public = 49	28.3 (8.0)	No = 54	27.7 (7.6)
	Total = 98	27.2 (8.9)	Total = 98	27.2 (8.9)	Total = 98	27.2 (8.9)
MPPUS—Total Score ^§^	M = 56	94.0 (14.6)	Private = 49	92.3 (17.8)	Yes = 44	90.8 (18.6)
	F = 42	90.3 (17.8)	Public = 49	92.5 (14.4)	No = 54	93.8 (13.8)
	Total = 98	92.4 (16.1)	Total = 98	92.4 (16.1)	Total = 98	92.4 (16.1)
MPPUS—F1 Abuse and Excessive Use	M = 56	40.8 (6.1)	Private = 49	40.8 (6.8)	Yes = 44	40.5 (6.8)
	F = 42	40.7 (6.7)	Public = 49	40.8 (5.8)	No = 54	40.9 (6.0)
	Total = 98	40.7 (6.3)	Total = 98	40.8 (6.3)	Total = 98	40.8 (6.3)
MPPUS—F2 Lack of Control	M = 56	29.0 (6.0)	Private = 49	28.6 (6.8)	Yes = 44	27.7 (7.1)
	F = 42	26.9 (6.8)	Public = 49	27.7 (6.1)	No = 54	28.4 (5.8)
	Total = 98	28.1 (6.4)	Total = 98	28.1 (6.4)	Total = 98	28.1 (6.4)
MPPUS—F3 Social Context-induced Craving	M = 56	24.2 (4.5)	Private = 49	23.0 (5.6)	Yes = 44	22.7 (5.9)
	F = 42	22.7 (5.8)	Public = 49	24.2 (4.6)	No = 54	24.4 (4.3)
	Total = 98	23.6 (5.1)	Total = 98	23.6 (5.1)	Total = 98	23.6 (5.1)

^§^ IAT—Internet Addiction Test; MMPIQ—Mobile Phone Involvement Questionnaire; MMPUS—Mobile Phone Problematic Use Scale; CEP—Coding Educational Program; trend to statistical significance (*p* = 0.078) due to a soft impact not evident in small samples (to be further studied).

**Table 5 ijerph-20-02983-t005:** Mean and standard deviation of the variables used to assess Emotional Dysregulation (DERS) and their differences by sex, school and participation in CEP.

Emotion Dysregulation Assessment	Sex		School		CEP	
DERS—Total Score ^§^	M = 56	78.2 (20.2) ^1^	Private = 48	81.9 (22.2)	Yes = 44	79.7 (22.2)
	F = 41	86.8 (24.2) ^1^	Public = 49	81.8 (22.5)	No = 54	83.5 (21.7)
	Total = 97	81.8 (22.2)	Total = 97	81.8 (22.2)	Total = 98	81.8 (22.2)
DERS—F1 Ego Dystonic Reactivity	M = 56	14.1 (4.1)	Private = 48	14.1 (4.2)	Yes = 44	14.4 (4.3)
	F = 41	14.9 (4.0)	Public = 49	14.8 (4.1)	No = 54	14.5 (4.0)
	Total = 97	14.1 (4.4)	Total = 97	14.1 (4.4)	Total = 98	14.1 (4.4)
DERS—F2 Attention Impairment	M = 56	13.0 (3.8) **	Private = 48	14.0 (4.6)	Yes = 44	13.9 (4.8)
	F = 41	15.6 (4.8) **	Public = 49	14.2 (4.4)	No = 54	14.3 (4.2)
	Total = 97	14.1 (4.5)	Total = 97	14.1 (4.5)	Total = 98	14.1 (4.5)
DERS—F3 Missing Strategies	M = 56	15.0 (6.0)	Private = 48	15.7 (6.4)	Yes = 44	15.4 (6.4)
	F = 41	16.5 (6.0)	Public = 49	15.5 (5.7)	No = 54	15.8 (5.8)
	Total = 97	15.6 (6.0)	Total = 97	15.6 (6.0)	Total = 98	15.6 (6.0)
DERS—F4 Impulsivity	M = 56	13.7 (5.9)	Private = 48	14.5 (6.5)	Yes = 44	14.0 (6.7)
	F = 41	15.0 (7.2)	Public = 49	14.0 (6.5)	No = 54	14.4 (6.3)
	Total = 97	14.2 (6.5)	Total = 97	14.2 (6.5)	Total = 98	14.2 (6.5)
DERS—F5 Reduced Emotional Recognition	M = 56	10.6 (3.7)	Private = 48	11.5 (4.8)	Yes = 44	10.3 (4.9) ^2^
	F = 41	12.0 (5.1)	Public = 49	10.9 (4.0)	No = 54	11.9 (3.8) ^2^
	Total = 97	11.2 (4.4)	Total = 97	11.2 (4.4)	Total = 98	11.2 (4.4)
DERS—F6 Reduced Self-Awareness	M = 56	5.7 (2.9)	Private = 48	5.3 (3.2)	Yes = 44	4.8 (3.3)
	F = 41	4.7 (3.1)	Public = 49	5.3 (3.0)	No = 54	5.6 (2.9)
	Total = 97	5.3 (3.1)	Total = 97	5.3 (3.1)	Total = 98	5.3 (3.1)

^§^ DERS—Difficulties in Emotion Regulation Scale; CEP—Coding Educational Program; ** Very significant sex effect (*p* < 0.01); ^1^ trend to statistical significance (*p* = 0.061) due to a soft impact not evident in small samples (to be further studied); ^2^ trend to statistical significance (*p* = 0.067) due to a soft impact not evident in small samples (to be further studied).

**Table 6 ijerph-20-02983-t006:** Mean and standard deviation of the variables used to assess Affective Dependence (DSRS) and their differences by sex, school and participation in Coding Educative Programs.

Affective Dependence Assessment	Sex		School		CEP	
DSRS—Total Score	M = 56	97.6 (12.2) ^1^	Private = 49	97.6 (11.8)	Yes = 44	98.1 (13.2)
	F = 42	102.4 (13.4) ^1^	Public = 49	101.8 (13.6)	No = 54	100.9 (12.6)
	Total = 98	99.7 (12.9)	Total = 98	99.7 (12.9)	Total = 98	99.7 (12.9)
DSRS—F1 Being Cared Self Identity	M = 56	5.6 (0.9) **	Private = 49	5.8 (1.0)	Yes = 44	5.9 (0.9)
	F = 42	6.2 (1.0) **	Public = 49	5.9 (1.0)	No = 54	5.8 (1.0)
	Total = 98	5.9 (1.0)	Total = 98	5.9 (1.0)	Total = 98	5.9 (1.0)
DSRS—F2 Lack of Self-Confidence	M = 56	−3.2 (0.8)	Private = 49	−3.2 (0.9)	Yes = 44	−3.3 ((0.9)
	F = 42	−3.2 (0.8)	Public = 49	−3.1 (0.7)	No = 54	−3.0 (0.7)
	Total = 98	−3.2 (0.8)	Total = 98	−3.2 (0.8)	Total = 98	−3.2 (0.8)
DSRS—F3 Complaisance	M = 56	1.6 (0.6)	Private = 49	1.6 (0.6)	Yes = 44	1.5 (0.7)
	F = 42	1.6 (0.7)	Public = 49	1.7 (0.7)	No = 54	1.7 (0.6)
	Total = 98	1.6 (0.7)	Total = 98	1.6 (0.7)	Total = 98	1.6 (0.7)
DSRS—F4 Self-Sufficiency	M = 56	3.6 (0.8)	Private = 49	3.8 (0.8) *	Yes = 44	3.7 (0.8) ^2^
	F = 42	3.5 (0.8)	Public = 49	3.4 (0.8) *	No = 54	3.4 (0.8) ^2^
	Total = 98	3.6 (0.8)	Total = 98	3.6 (0.8)	Total = 98	3.6 (0.8)

DSRS—Dependence Self-Rating Scale; CEP—Coding Educational Program; * Significant school effect (*p* < 0.05); ** Very significant sex effect (*p* < 0.01); ^1^ trend to statistical significance (*p* = 0.066) due to a soft impact not evident in small samples (to be further studied); ^2^ trend to statistical significance (*p* = 0.084) due to a soft impact not evident in small samples (to be further studied).

**Table 7 ijerph-20-02983-t007:** Factorial analysis of Affective Dysregulation (DERS) and Affective Dependence (DSRS) subscores.

Original Parameters	FA1 Emotional Dysregulation 36.7% ^1^	FA2 Reduced Self-Confidence 17.4%	FA3 Self-Sufficiency 14.7%
DERS—F4 Impulsivity	0.840		
DERS—F2 Attention Impairment	0.834		
DERS—F3 Missing Strategies	0.796		
DERS—F1 Ego Dystonic Reactivity	0.730		
DERS—F5 Reduced Emotion Recognition	0.678		
DSRS—F1 Being Cared Self Identity	0.619	−0.585	
DERS—F6 Reduced Self-Awareness		0.794	
DSRS—F2 Lack of Self-Confidence	0.390	0.664	
DSRS—F3 Complaisance			−0.848
DSRS—F4 Self-Sufficiency			0.732

^1^ Explained variance after varimax rotation; Values lower than 0.350 are not shown.

**Table 8 ijerph-20-02983-t008:** Correlations between original scales’ Total Scores and Extracted Factors.

Original Parameters		FA1 Emotional Dysregulation	FA2 Reduced Self-Confidence	FA3 Self-Sufficiency
DSRS—Total Score	Pearson r	0.647 ***	−0.169	−0.477 ***
	2-tails *p*	<0.001	0.097	<0.001
	N	97	97	97
DERS—Total Score	Pearson r	0.968 ***	0.125	0.103
	2-tails *p*	<0.001	0.224	0.316
	N	97	97	97

*** Statistical Difference significant at *p* < 0.001.

**Table 9 ijerph-20-02983-t009:** Factorial analysis of variables used to assess behavior associated with DMPU.

Original Parameters	FB1: DMPU Problematic Use 39.5% ^1^	FB2: Night Use 20.2% ^1^	FB3: Day Use 20.1% ^1^
MPPUS—F2 Lack of Control	0.892		
MPPUS—F3 Social Context-induced Craving	0.857		
MPPUS—F1 Abuse and Excessive Use	0.843		
IAT—Total Score	0.820		
MPIQ—Total Score	0.786		
Night Use in whole Weekend		0.946	
Night Use in whole Working Days		0.916	
Day Use in whole Working Days			0.921
Day Use in whole Weekend			0.904

^1^ Explained variance after varimax rotation.

**Table 10 ijerph-20-02983-t010:** Correlations between the extracted factors and the total scores of IAT, MMPIQ, MPPUS and the time schedule of mobile phone use.

Original Scale or Parameter		FB1: DMPU Problematic Use	FB2: Night Use	FB3: Day Use
IAT—Total Score	Pearson r	0.820 ***	0.064	0.045
	2-tails *p*	<0.001	0.536	0.660
	N	97	97	97
MPIQ—Total Score	Pearson r	0.786 ***	0.106	0.226 *
	2-tails *p*	<0.001	0.301	0.026
	N	97	97	97
MPPUS—Total Score	Pearson r	0.960 ***	0.037	0.009
	2-tails *p*	<0.001	0.719	0.933
	N	97	97	97
Day Use in whole Working Days	Pearson r	0.021	−0.169	0.921 ***
	2-tails *p*	0.835	0.098	<0.001
	N	97	97	97
Day Use in whole Weekend	Pearson r	0.106	0.157	0.904 ***
	2-tails *p*	0.301	0.125	<0.001
	N	97	97	97
Night Use in whole Working Days	Pearson r	0.080	0.916 ***	0.248 *
	2-tails *p*	0.438	<0.001	0.014
	N	97	97	97
Night Use in whole Weekend	Pearson r	0.075	0.946 ***	0.100
	2-tails *p*	0.465	<0.001	0.332
	N	97	97	97

*** Statistical difference significant at *p* < 0.001; * Statistical difference significant at *p* < 0.05.

**Table 11 ijerph-20-02983-t011:** Correlations between Psychological Factors and Behavioral Factors.

Original Parameters		FB1: DMPU Problematic Use			FB2: Night Use			FB3: Day Use		
		All	Coding		All	Coding		All	Coding	
			Yes	Not		Yes	Not		Yes	No
FA1 Emotional Dysregulation	Pearson r	0.574 ***	0.605 ***	0.572 ***	0.093	0.176	0.027	−0.130	−204	−115
	*p*	<0.001	<0.001	<0.001	0.363	0.258	0.849	0.206	0.190	0.409
	N	97	43	54	97	43	54	97	43	54
FA2 Reduced Self-Confidence	Pearson r	0.023	0.027	0.064	0.037	−0.032	0.184	−0.073	−0.097	−0.118
	*p*	0.823	0.863	0.646	0.722	0.837	0.183	0.475	0.537	−396
	N	97	43	54	97	43	54	97	43	54
FA3 Self-Sufficiency	Pearson r	0.033	0.117	−0.038	0.320 **	0.279	0.334 *	0.105	−0.034	0.262
	*p*	0.750	0.455	0.788	0.001	0.070	0.014	0.304	0.827	0.055
	N	97	43	54	97	43	54	97	43	54

*** Statistical difference significant at *p* < 0.001; ** Statistical difference significant at *p* < 0.01; * Statistical difference significant at *p* < 0.05.

**Table 12 ijerph-20-02983-t012:** Regression analysis on extracted factors to predict the level of smartphone problematic use.

		Non-Standardized Coefficients		Standardized Coefficients		
	Model	B	Error DS	Beta	t	Sig.
All	(Constant)	0.000	0.084		0.000	1.000
	FA1—Dysregulation	0.574	0.084	0.574	6835	<0.001
Yes CEP	(Constant)	0.135	0.146		0.925	0.360
	FA1—Dysregulation	0.704	0.144	0.605	4.872	<0.001
No CEP	(Constant)	0.093	0.094		−0.989	0.327
	FA1—Dysregulation	0.480	0.096	0.572	5.025	0.000

**Table 13 ijerph-20-02983-t013:** Discriminant analysis on behavioral parameters by Coding Educational Program attendance.

Parameter	Stand Coefficient	Wilks Lambda	X^2^	df	Sig.
Day Use in whole Working Days ^1^	−1.521	0.904	9.500	2	0.009
Day Use in whole Weekend ^1^	1.069				
Group	Centroids	Correct Classifications			
Yes CEP	0.362	63.9%			
No CEP	−288				

^1^ Total hours in the whole period.

**Table 14 ijerph-20-02983-t014:** Association between attendance of Coding Educational Programs and type of activity for which the mobile phone is used.

Mobile Phone Use			Attended CEP		Total
			Yes	No	
To obtain information ^1^	No	Count	16	34	50
		Percent	39.0%	64.2%	53.2%
	Yes	Count	25	19	44
		Percent	61.0%	35.8%	46.8%
Total		Count	41	53	94
		Percent	100.0%	100.0%	100.0%
To improve orientation ^2^	No	Count	31	51	82
		Percent	75.6%	96.2%	87.2%
	Yes	Count	10	2	12
		Percent	24.4%	3.8%	12.8%
Total		Count	41	53	94
		Percent	100.0%	100.0%	100.0%

^1^ X^2^ = 5.862; *p* = 0.013; ^2^ X^2^ = 8.823; *p* = 0.004.

## Data Availability

The datasets for this study are available from the corresponding author on reasonable request.

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
