# Peer review of "Impact of Coding Educational Programs (CEP) on Digital Media Problematic Use (DMPU) and on Its Relationship with Psychological Dependence and Emotional Dysregulation"

_ijerph, 2023, doi:10.3390/ijerph20042983_

Round 1
Reviewer 1 Report
The paper is of a very good quality. Its only drawback I see is its orientation to students from one city only. This reduces the applicability of researchers' findings to a wider population.
Author Response
Thank you for all your precious suggestions and for your appreciation.
We are in complete agreement with you about the need to extend our observation to wider population taken from different cities and schools. This is a pilot study.
Best regards
The authors

Reviewer 2 Report
First of all, I thank the editor for allowing me to review the paper which is very interesting, full of data and methods and well written by the authors. The subject is well introduced through sufficient references to the literature. The analysis tools are well described and their validation is mostly present in the bibliography. The discussion is also consistent with the results obtained and proposes an advance in the knowledge of a very relevant phenomenon which has recently had a notable increase among the youngest.
Only two things, not related to the drafting of the paper, can be indicated to the authors:
1 - such a well-organized research should have had more observations in the field because the weakness of the analyzes reported can be traced in the low number especially within the subgroups of interest; in fact, just as an example, even a simple analysis of independence through the chi-square test cannot be performed if it relates to a number of cases lower than 5 and the performance of the multivariate analysis methods is greatly affected by a very small number of cases; finally, the presence of researchers during the survey can be very important to prevent distortions due to suggestions from peers or from the supervisor which, in the case of few observations, can be decisive in leading to incorrect results due to poor data quality.
2 - probably this kind of experiments, even when the anonymity of the respondents who are protected from the point of view of privacy is observed to the maximum, would deserve an ethical evaluation because the impact on the minors involved must be taken into consideration in the research protocol. Is it possible to have a reference to an ethical evaluation or at least to an analysis consideration of the research protocol which is related to the protection of the interested parties from an ethical point of view?
Author Response
Thank you for all your precious suggestions and for your appreciation.
Best regards
The authors
